# Establishing a protocol for the compatibilities of closed-system transfer devices with multiple chemotherapy drugs under simulated clinical conditions

**Shao-Chin Chiang** [1,2]*, **Mandy Shen**[3], **Chen-Chia Lin**[1], **Hui-Ping Chang**[4]

1 Department of Pharmacy, Koo Foundation Sun Yat-Sen Cancer Center, Taipei, Taiwan, 2 Department of Pharmacy, School of Pharmaceutical Sciences, National Yang Ming Chiao Tung University, Taipei, Taiwan, 3 Department of Biochemical Science and Technology, National Taiwan University, Taipei, Taiwan, 4 College of Nursing, Taipei Medical University, Taipei, Taiwan

* scchiang316@nycu.edu.tw

**Data Availability Statement:** All relevant data are within the manuscript and its Supporting Information files.

## Abstract

Closed-system drug transfer devices (CSTDs) are used to prevent occupational exposure to hazardous drugs in health care providers. They are considered Class II medical devices by the US FDA and are cleared but not approved before marketing. While compatibility tests are conducted by CSTD manufacturers, the procuring institution needs to consider performing its own studies before buying these devices. Herein we tested the compatibility of the components of the Needleless® DualGuard CSTD system (vial access clips, vial access spikes, and administration adaptors) with 10 antineoplastic drugs, under simulated clinical conditions, including compounding and administration, and examined drug potency maintenance, plasticizer migration, and device functionality. All drugs maintained potency within 5%. Diisononyl phthalate leakage was observed from the administration adaptors for paclitaxel and concentrated etoposide solution. In addition, white particles were discovered in CSTDs storing busulfan solution and small cracks were observed on devices which stored melphalan. Thus, it was concluded that even in simulated clinical conditions, instead of extreme conditions, there are still concerns regarding the efficacy and safety of CSTD components. The methodology may be used to implement and detect possible interactions between antineoplastic agents and CSTD components before procurement.

## Introduction

Closed-system drug transfer devices (CSTDs) are designed to ensure that hazardous drugs (HDs) are handled in an airtight and leakproof manner [1]. Several peer-reviewed studies have demonstrated effective control of surface contamination and reduction in personnel exposure with the use of CSTDs [2–4]. Since the release of the US Pharmacopeia (USP) Chapter <800>, the regulatory board has mandated the use of CSTDs for the administration of HD and has recommended their use for preparation of HD [1]. Simultaneously, US FDA 510(k) clearance

**Funding:** This work was supported by Needleless Corporation (http://www.needleless-medical.com/en), which funded all costs associated with antineoplastic drugs and device purchase. The funders had no role in study design, data collection and analysis, decision to publish, or preparation of the manuscript. No grant number was attached to this study.

**Competing interests:** This work was supported by Needleless Corporation (Taiwan) (http://www.needleless-medical.com/en). We know of no conflict of interest associated that would influence the outcome of this publication. The funder had no role in the study design, data collection and analysis, and the decision to publish this manuscript. No grant number was attached to this study. This does not alter our adherence to PLOS ONE policies on sharing data and materials.

has categorized CSTDs as meeting the US National Institute for Occupational Safety and Health definitions, under the ONB product code [1]. However, the ONB product code does not entail any performance standards. Therefore, it is inevitable for healthcare institutions to evaluate CSTDs in accordance with the manufacturers' claims, e.g., airtightness, no plasticizer leakage, no interactions with tested drugs, etc.

Recent studies have reported incompatibilities related to CSTDs, which include plastic breakage, drug adsorption, and plasticizer contamination [5–10]. Plastic breakage can cause drug leakage, which increases HD exposure [2, 11]. Drug adsorption by medical devices results in the transfer of a lower dose to the patient [5, 7]. Moreover, plasticizers such as phthalates can migrate into drug admixtures, and when administered to patients, can induce reproductive toxicity, carcinogenesis, cardiotoxicity, hepatotoxicity, and nephrotoxicity; this can negatively affect patients' health [7–10, 12–14]. Although compatibility tests are routinely performed by the manufacturers of CSTDs and no negative data are reported, incompatibility incidents continue to occur. Multiple studies have shown that the incompatibility concerns are related to the solvents in the formulations containing antineoplastic drugs and protein drugs [8–10, 14, 15]. Therefore, testing of the bulk powder form of antineoplastic drugs is insufficient, and individual formulations of the target drugs should be tested. Discrepancies in the manufacturers' protocols may cause difficulties for the procuring institutions while comparing products and may fail to reflect actual clinical usage. For this reason, it is crucial to develop a compatibility protocol that incorporates clinicians' perspectives and is feasible for clinicians to implement when required, especially when a new drug has been released and has not been tested by any CSTD manufacturers.

This study proposes a methodology to test CSTD (DualGuard CSTD system) compatibility with commercial formulations of antineoplastic drugs containing potentially incompatible solvents. All drug manipulations and experimental conditions were designed to mimic clinical usage, including concentrations for clinical use and exposure time. The compatibility of CSTDs with the antineoplastic drugs was examined with respect to drug potency maintenance, plasticizer migration, and device functionality.

## Materials and methods

### Materials

Antineoplastic drugs were selected based on the following two criteria: (i) commonly used chemotherapy drugs, classified by their mechanism of action and (ii) drugs that pose potential incompatibility risks. Busulfan solution (Baxter Inc., Westfalen, Germany) contains N, N-dimethylacetamide (DMA), as per a US FDA warning [2], which can dissolve polycarbonate and ABS [2, 16, 17]. Etoposide solution (Fresenius Kabi, Himachal Pradesh, India) was reported to cause di (2-ethylhexyl) phthalate (DEHP) leakage, because it exists as a dehydrated alcohol formulation and due to the lipophilic nature of one of its components, polysorbate 80 [7]. Similarly, paclitaxel solution (Fresenius Kabi, Himachal Pradesh, India) which contains polyoxyl 35 castor oil and dehydrated alcohol, and melphalan solution (GlaxoSmithKline plc., Strada, Italy), which contains concentrated dehydrated ethanol (96%), were added to the list due to DEHP leakage risk [8, 9]. Cisplatin (Fresenius Kabi, Himachal Pradesh, India) was chosen for its wide application in chemotherapy. Cyclophosphamide (Baxter Inc., Halle-Kunsebeck, Germany), fluorouracil (Haupt Pharma GmbH., Wolfratshausen, Germany), irinotecan (Fresenius Kabi, Himachal Pradesh, India), doxorubicin (Pfizer Inc., Western Australia, Australia), and vinorelbine (Fresenius Kabi, Himachal Pradesh, India) were chosen because they are the most commonly used drugs in their respective therapeutic classes. Commercial 0.9% sodium chloride was obtained in 100-, 250-, and 500-mL non-polyvinyl chloride containers

(Nang Kuang Pharma. Co., Ltd., Taipei, Taiwan), and 500 mL of 5% dextrose solution (in water) was obtained in glass bottles (Vitagen, Taiwan Biotech Co., Ltd, Taoyuan, Taiwan). Syringes made of polypropylene materials were obtained (Terumo, Tokyo, Japan). All drugs were tested in their final product forms. At the same time, we also provide information on the polarity and pH of the antineoplastic drugs in the S1 Table.

The solution for HPLC formic acid (purity >95%), acetonitrile, and methanol were purchased from Sigma-Aldrich. Ethyl acetate was obtained from J.T. Baker (Taipei, Taiwan). Quality control BPA (99.9%) was purchased from Sigma-Aldrich (Taipei, Taiwan). Custom phthalate standards for BBP, DBP, DEHP, DIDP, DINP, and DNOP (2000 ppm) were obtained from AccuStandard (Taipei, Taiwan). An additional DEHP-D4 (99.6%) internal standard was included, purchased from Sigma-Aldrich (Taipei, Taiwan). The mobile phase and extraction solvents used were acetonitrile (99.5%; Chem Service, Taipei, Taiwan), ammonium acetate (98.7%; Honeywell, Taipei, Taiwan), and methanol (100%; Merck, Taipei, Taiwan).

## DualGuard CSTD system

The DualGuard CSTD system (Needleless Corp., Taoyuan, Taiwan) includes closed male luer, needleless connectors, vial access spikes, intravenous burette sets, and bag spikes made of acrylonitrile butadiene styrene (ABS) and polycarbonate. For each drug, three CSTD device components were tested in triplicate. The components included vial access clips (Fig 1A), vial access spikes (Fig 1B), and administration sets (Fig 1C).

## Preparation of antineoplastic drug solution

In order to simulate the clinical conditions, all the components in the preparations of antineoplastic drugs followed the usual clinical practices, i.e. using commercial drug products, clinically prescribed doses, and standardized dilution according to the institutional guideline. All drug solutions were prepared by an experienced compounding pharmacist, under aseptic conditions in a Class II B, laminar-airflow biologic safety cabinet. Antineoplastic drugs, in final product forms, were reconstituted as per the highest dose level suggested in the package inserts, and the administration dosage was calculated using a typical adult body surface area ($1.7 \text{ mg/m}^2$) and weight (70 kg). Final concentrations were determined using the highest stable concentrations recorded in the *Handbook on Injectable Drugs* [18] (S2 Table). For both the vial access clip and vial access spike, half of the total vial dosage was used, which simulated the condition of the leftover. The compounding methods are detailed in S2 Table. Etoposide was prepared in low (170 mg, 0.4 mg/mL) and high (1275 mg, 12 mg/mL) concentrations in 0.9% sodium chloride due to its special formulation and clinical uses.

## Study design for drug potency maintenance and plastic migration tests

To determine the effect of storage of leftover drugs after compounding in CSTDs, we analyzed drug potency and drug compatibility after preparation. Stability information was obtained from package inserts and the *Handbook on Injectable Drugs* [18]. All vial samples were collected after 8 h (the average vial storage time after compounding for multiple-dose vials) for testing both potency maintenance and plasticizer migration, except for melphalan. Melphalan was left at 25°C for 90 min before the potency maintenance test and 8 h for the plasticizer migration test. Drug vials were shaken for 1 min every hour during storage, to increase drug-device direct contact time. Vial access clips were not used for vinorelbine because the vial size was unsuitable for the device. Blank 0.9% sodium chloride and 5% dextrose (in water) were

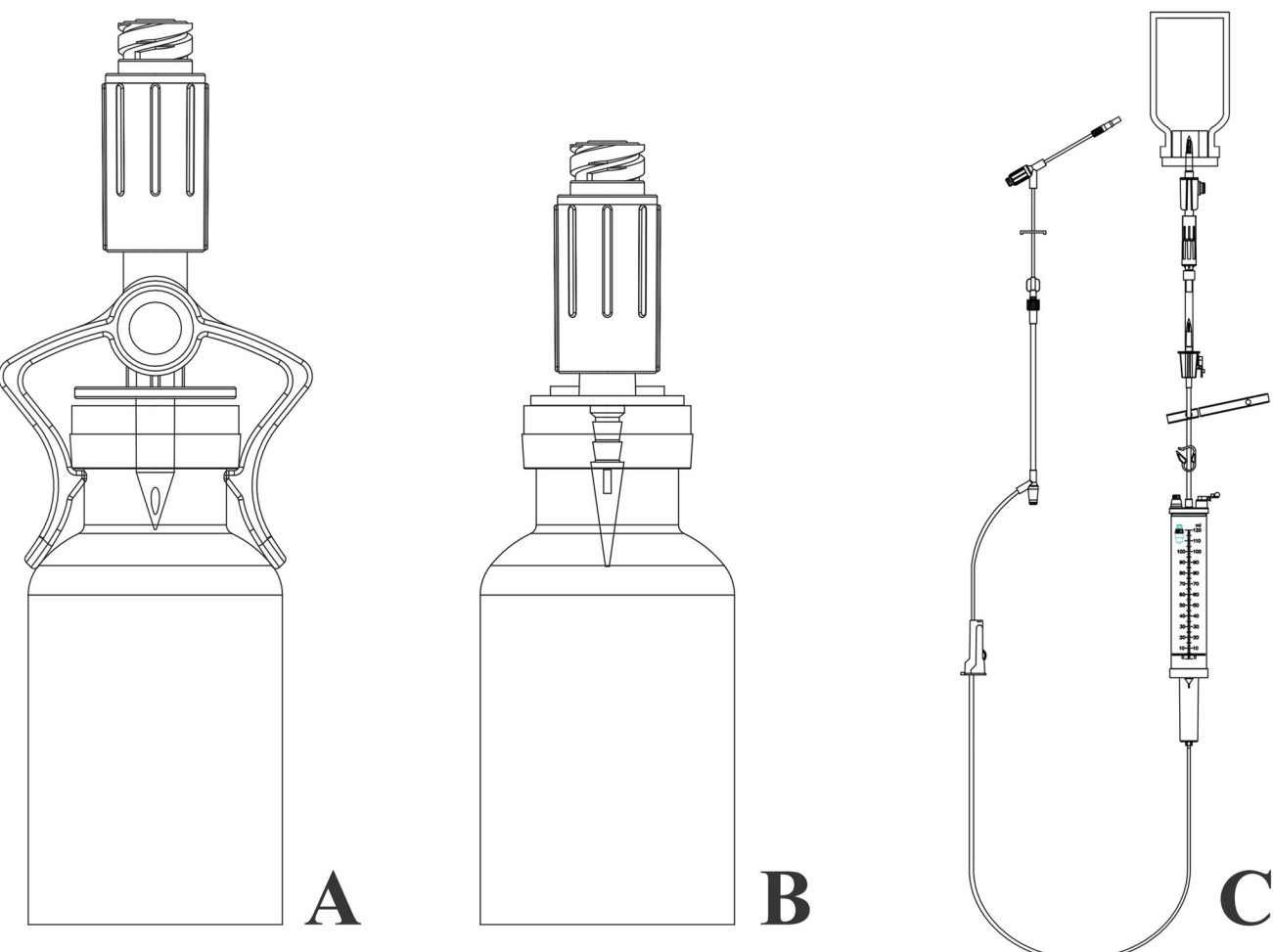

**Fig 1. Illustrations of drug preparation and administration device assembly.** (A) Vial access clip with *DualGuard* needleless connector attached to a drug vial. (B) Vial access device with *DualGuard* needleless connector attached to a drug vial. (C) Administration set with closed male luer and *DualGuard* needleless connector attached to an intravenous bag.

stored with vial access clips and vial access spikes, at room temperature for 8 h as controls. Table 1 shows the vial storage conditions and durations.

In the case of the administration set, the intravenous bags with drug solutions were spiked with administration sets, and fluid was allowed to flow through the sets. All test infusions were performed at ambient temperature. The administration information was obtained from the package inserts and the *Handbook on Injectable Drugs* [18] and adjusted according to the guidelines of the Koo Foundation Sun Yat-Sen Cancer Center. In addition, the compounded sterile preparations were assigned to different risk levels, according to the 2008 USP797 [19] guidelines which classify formulations into three levels of risk: low, medium, and high, depending on the compounding environment, whether the ingredients are sterile, and the complexity of the operating procedures; different Beyond-Use Dates (BUD) were assigned according to the risk level. The majority of antineoplastic drug preparations were classified as medium risk. The maximum BUD was 30 h or less (12 h if the secondary engineering control could not reach the ISO Class 7 standard). Therefore, the maximum infusion time was set as 24 h. Samples for the tests on drug potency maintenance were collected after the predetermined clinical infusion time, while samples for plasticizer migration analysis were collected

**Table 1. Storage durations and conditions of the reconstituted vials and diluted infusion bags before the sampling for potency maintenance and plasticizer migration tests.**

| Drugs | Vial access clip | | Vial access device | | Administration set | |
|---|---|---|---|---|---|---|
| | Potency | Plasticizer migration | Potency | Plasticizer migration | Potency | Plasticizer migration |
| Busulfan | 8 h at 2–8˚C | 8 h at 2–8˚C | 8 h at 2–8˚C | 8 h at 2–8˚C | 2 h | 2 h |
| Etoposide | 8 h at 25˚C | 8 h at 25˚C | 8 h at 25˚C | 8 h at 25˚C | 1 h | 2 h |
| Paclitaxel | 8 h at 25˚C | 8 h at 25˚C | 8 h at 25˚C | 8 h at 25˚C | 3 h | 7.6 h |
| Melphalan | 1.5 h at 25˚C | 8 h at 25˚C | 1.5 h at 25˚C | 8 h at 25˚C | 20 min | 40 min |
| Cisplatin | 8 h at 25˚C | 8 h at 25˚C | 8 h at 25˚C | 8 h at 25˚C | 320 min | 320 min |
| Cyclophosphamide | 8 h at 2–8˚C | 8 h at 2–8˚C | 8 h at 2–8˚C | 8 h at 2–8˚C | 4 h | 4 h |
| Fluorouracil | 8 h at 25˚C | 8 h at 25˚C | 8 h at 25˚C | 8 h at 25˚C | 250 min | 250 min |
| Irinotecan | 8 h at 2–8˚C | 8 h at 2–8˚C | 8 h at 2–8˚C | 8 h at 2–8˚C | 1.5 h | 3 h |
| Doxorubicin | 8 h at 2–8˚C | 8 h at 2–8˚C | 8 h at 2–8˚C | 8 h at 2–8˚C | 6 h | 6 h |
| Vinorelbine | NT | NT | 8 h at 2–8˚C | 8 h at 2–8˚C | 30 min | 2 h |

NT, not tested.

after twice the predetermined clinical infusion time or at the end of the infusion (Table 1). Blank 0.9% sodium chloride and 5% dextrose (in water) were allowed to flow for 24 h before analysis, as controls.

**Drug potency maintenance test parameters and data analysis.** All samples were analyzed by the Société Générale de Surveillance (SGS, Taiwan). Drug potency maintenance was evaluated with peak area percentage change as an indicator of concentration loss. We set a limit of 10% peak area change because all drug preparations must maintain less than a 10% decrease in concentration to be considered stable [18, 20]. The peak area immediately after preparation (initial sample, $t_i$) was defined as 100%, and peak area after storage or infusion (final sample, $t_f$) was compared with the $t_i$ peak area to compute percentage change. Variability in drug content was evaluated with the relative standard deviation (RSD, %) calculated from the composite peak areas of three replicates. We set a maximum limit of <2% for RSD, as per the valid analytical range [20, 21]. Samples were analyzed by high performance liquid chromatography with a diode-array detection (HPLC-DAD) system, equipped with an Agilent 1200 series HPLC system (MN, USA) and Agilent Eclipse XDB-C8 column (150 × 4.6 mm, 5 μm particle size). The HPLC-DAD protocol was adapted from Poujol et al. [22]. Drug HPLC assays were validated for the linearity of concentration and response curve. Conditions for each active ingredient analysis are outlined in S3 Table. Due to technical difficulties, busulfan was analyzed through gas chromatography-mass spectrometry (GC-MS). The analysis was performed using an Agilent Technologies 6890 series GC (Taipei, Taiwan), equipped with a 5973 series MS. The column consisted of a 30-m phenyl arylene polymer Agilent J&W DB-5MS capillary column (Taipei, Taiwan) with an internal diameter of 0.25 mm and a film thickness of 0.25 μm. The carrier helium gas was set at a flow rate of 1 mL/min. The injector temperature was maintained at 280˚C. The oven temperature was initially set at 80˚C and slowly increased at 25˚C/min to 200˚C for 6 minutes. The MS baking temperature was 230˚C for the source and 150˚C for the quadrupole with a transfer line temperature of 280˚C. The MS scanned 35 to 350 m/z for busulfan ion, which had a retention time of 2 min.

**Plasticizer migration test parameters and data analysis.** A plasticizer migration test attempted to detect bisphenol A (BPA) and six common phthalates found in medical devices, namely butyl benzyl phthalate (BBP), di-n-butyl phthalate (DBP), di(2-ethylhexyl) phthalate (DEHP), di-n-octyl phthalate (DNOP), diisononyl phthalate (DINP), and diisodecyl phthalate

(DIDP). BPA calibration standards were prepared with a series of acetonitrile dilutions. A six-point calibration curve was established, covering a range of 50–500 ng/mL. Custom phthalate calibration standards were prepared using a series of methanol dilutions, with concentrations ranging from 50 to 500 ng/mL. An aliquot of 990 μL from each phthalate standard was mixed with 10 μL of internal DEHP-D4 standard stock solution (24.9 ppm) to prepare a calibration curve. The concentration of the internal standard at each calibration level was 249 ng/mL. The concentration of BPA and six phthalates were verified using the linear calibration curves.

The LC-MS/MS analysis for BPA was performed using an API 4000 system (Applied Biosystems), with an Agilent 1260 series HPLC system (MN, USA). Phthalate was analyzed using an API 3200 System (Applied Biosystems), with an Agilent 1200 series HPLC-DAD system. Both HPLC systems were equipped with an Agilent Eclipse XDB-C8 column (150 × 2.1 mm, 3.5 μm particle size). For the BPA assay, the test solution was first diluted 20 times with acetonitrile. The mobile phase comprised an equal mixture of deionized water and acetonitrile. The injection volume was 5 μL, and the flow rate was 0.4 mL/min. Mass spectrometry (MS) was performed in negative electrospray ionization (ESI) mode with a capillary voltage of −4500 kV, nitrogen gas (40 L/h), and a desolvation temperature of 450˚C. Multiple reaction monitoring was performed, and BPA ions were scanned at 93.1, 133, and 212 m/z. Conversely, the phthalate assay comprised a gradient mobile phase, prepared from a mixture of water and 5 mM ammonium acetate in methanol. The test solution was diluted 20 times with acetonitrile before loading. The injection volume was 10 μL, and the flow rate was 0.4 mL/min. MS was performed under APCI-positive ESI with a capillary voltage of 5500 V, nitrogen gas (50 L/h and 55 L/h), and a desolvation temperature of 500˚C. Multiple reaction monitoring was performed, which detected DEHP-D4 at 153 m/z, BBP at 149, 205, and 239 m/z, DBP at 149, 205, and 121 m/z, DEHP at 149, 167, and 279 m/z, DIDP at 149, 289, and 307 m/z, DINP at 149, 275, and 293 m/z, and DNOP at 149, 261, and 121 m/z. The concentration of the compounds was calculated using the formula: concentration (ng/mL) × dilution factor × volume (mL).

### DualGuard CSTD functionality test

Functionality testing followed the ISO8536-4:2000 guidelines, and was divided into four sections: (1) visual observations, (2) air leakage, (3) spiking, and (4) a tensile test. Device breakage, erosion, or any appearance abnormalities were examined visually. In the air leakage test, the devices were submerged in water, followed by a supply of more than 20 kPa of air for 15 s, and then inspected for air bubbles. The spiking test detected vial scoring after repetitive access. In the tensile test, 11 specific spots on administration sets were examined as shown in Fig 2. Because the joints were relatively fragile, we selected each joint for the functionality tensile test. An administration set was clamped onto the tensile testing machine (model number QC-508M20) and pulled at a speed of 200 mm/min. The acceptance was set to withstand 15 N for 15 s.

## Results

Results from the potency maintenance test revealed that all drug was not found to exceed the 10% threshold (Table 2).

In the plasticizer migration test, while no plasticizer leakage was detected in the 0.4 mg/mL etoposide samples, significant DINP leakage (an average of 900000 mg/kg) was observed in high-concentration etoposide infusion solution (1.2 mg/mL). In addition, DINP contamination (average: 20 mg/kg) was detected in paclitaxel samples in administration sets after 7.6 h of simulated infusion.

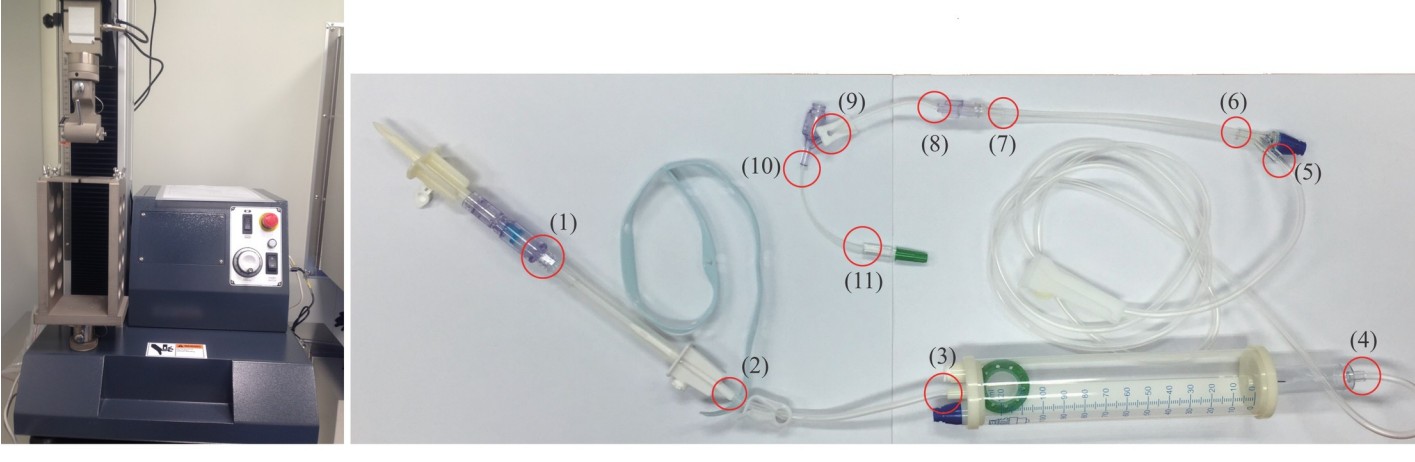

**Fig 2. Detection points on administration sets for the tensile test.** The red circles are the detection points.

Results from the functionality testing showed the presence of white particles on the vial access spikes after exposure to busulfan. However, a closer examination under a microscope (magnification: 30×) revealed no breakage of device (Fig 3A). Besides the detection of white particles, air leakage and spiking tests did not reveal other indications of abnormalities. After storing the melphalan solution for 8 h, cracks were found on the syringe attachment end on vial access spikes (Fig 3B) and administration sets (Fig 3C), but not on the vial access clip. Functionality tests for other drugs passed the visual assessment, air leakage, and spiking tests. Tensile tests were performed on administration sets, and all drugs could withstand >15 N (approximately 17 N) for 15 seconds, similar to control samples (0.9% sodium chloride), which could endure 17 N.

## Discussion

To date, no standard criteria have been developed to accurately ascertain CSTD compatibility. One essential consideration for a CSTD compatibility test is to test the device with the final

**Table 2. Results of drug potency maintenance tests for all CSTD components.**

| Drugs | Peak area % change[a] | | |
| --- | --- | --- | --- |
|  | Vial access clip | Vial access device | Administration set |
| Busulfan | 1.99% ± 1.00 | 0.79% ± 0.53 | 4.63% ± 2.29 |
| Etoposide (low) | 0.48% ± 0.20 | 0.58% ± 0.15 | 0.45% ± 0.18 |
| Etoposide (high) | 0.79% ± 0.59 | 1.31% ± 1.37 | 0.85% ± 0.65 |
| Paclitaxel | 0.33% ± 0.08 | 0.10% ± 0.07 | 0.81% ± 0.10 |
| Melphalan | 0.01% ± 0.01 | 0.06% ± 0.02 | 0.12% ± 0.13 |
| Cisplatin | 0.08% ± 0.04 | 0.04% ± 0.03 | 0.05% ± 0.05 |
| Cyclophosphamide | 0.06% ± 0.05 | 0.08% ± 0.02 | 0.27% ± 0.14 |
| Fluorouracil | 0.02% ± 0.02 | 0.04% ± 0.03 | 0.04% ± 0.00 |
| Irinotecan | 0.06% ± 0.08 | 0.05% ± 0.05 | 0.19% ± 0.06 |
| Doxorubicin | 0.03% ± 0.02 | 0.07% ± 0.03 | 0.09% ± 0.06 |
| Vinorelbine | NT | 0.06% ± 0.04 | 0.21% ± 0.06 |

[a]Peak area % change was calculated by subtracting the absolute peak area value of the before sample from that of the after sample, followed by division by the peak area of the before sample.

NT, not tested.

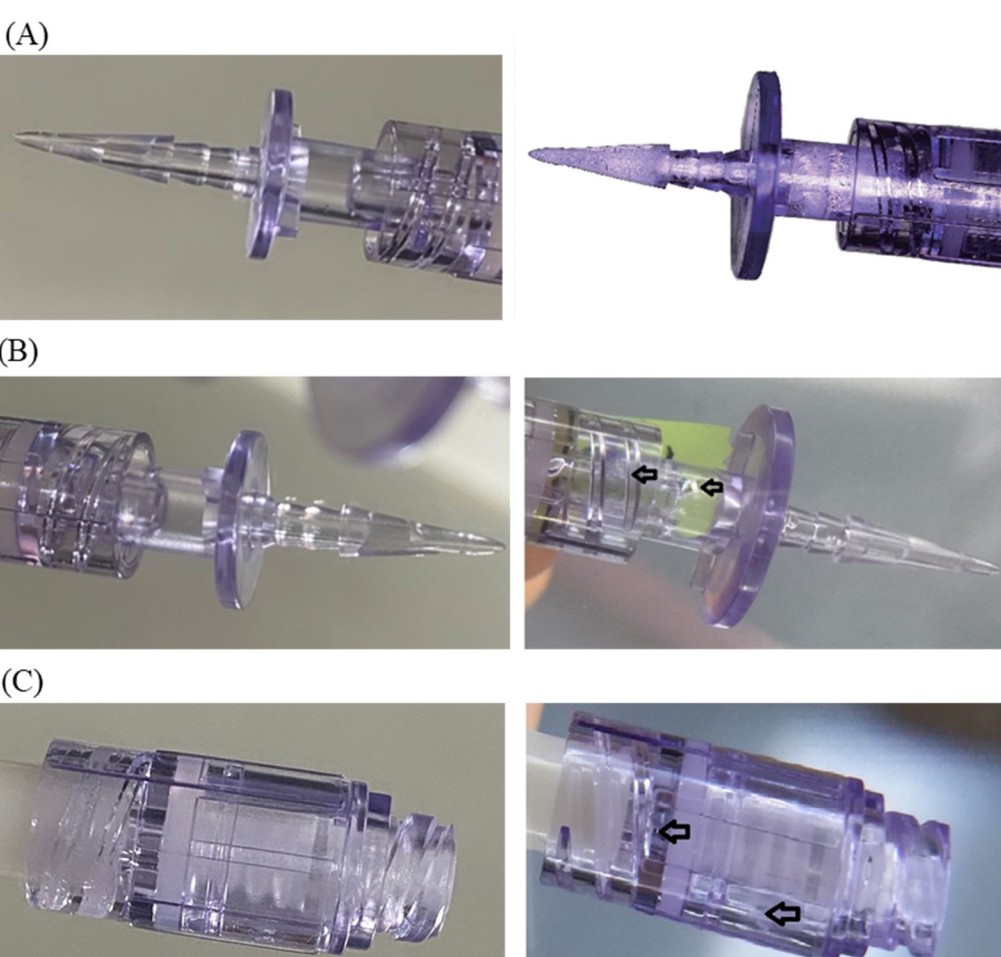

**Fig 3. Changes in the appearance of CSTDs during functionality tests.** (A) White matter appeared on the vial access device after 8 h of storage with busulfan (photo shown on the right). New vial access device as control (photo shown on the left). (B) The vial access connector was found to have cracks after being exposed to melphalan for 8 h with vial accessed device (photo shown on the right). New vial access device as control (photo shown on the left). (C) The needleless connector was found to have cracks after being exposed to melphalan for 8 h with administration set (photo shown on the right). The new needleless connectors were used as control (photo shown on the left).

drug product. Herein, we designed a compatibility test of CSTDs and antineoplastic drugs, based on clinicians' perspectives. We simulated the usage of the tested formulations in clinical practices by using clinical concentrations, durations, and conditions. Subsequently, we analyzed the effects of drug-device interactions through drug potency maintenance, plasticizer migration, and device functionality tests. Simultaneously, the multiple studies have demonstrated that solvents contribute to plasticizer extraction and plastic breakage [2, 6, 7, 16, 18]. Examples of solvents with potential risks are DMA, N-methylformamide, Cremophor/polysorbate 80, and absolute alcohol [2, 7–10, 16]. Took the above factors into consideration and among the drugs that our cancer center currently uses, we selected paclitaxel, melphalan, busulfan, and etoposide as representative drugs to reflect the real situation of clinical use.

Busulfan is known to be incompatible with polycarbonate and ABS materials [18]. In our study, we observed the formation of white particles on the vial access spikes (polycarbonate in the fluid pathway) after exposure to busulfan solution, while no precipitate was observed on the vial access clips (ABS in the fluid pathway). These results were in line with the

incompatibility reports suggested in the *Handbook of Injectable Drugs* and warnings [2, 16–18]. Moreover, melphalan caused the syringe attachment end of the vial access spikes to break. We did not observe any precipitate on devices with etoposide, although the literature suggests that etoposide can react with ABS materials [18]. We could not eliminate the possibility of incompatibility because the vial access clip (ABS in the fluid pathway) is white, which might make it difficult to detect precipitates. The causes of the white matter formation remain unknown. Future studies are warranted to investigate the derivatives found on the devices or drug solution through interactions between DMA and CSTD materials. This may yield a new requirement for device-drug compatibility. In addition, etoposide solution was found to extract a significant amount of plasticizers (DINP) from the administration set, as anticipated [18]. Similar results were also observed in the paclitaxel solution [8, 9, 18].

In order to compare that the method established in this study for compatibility testing is more suitable for clinical use, we compared with two other compatibility studies, performed by two different CSTD manufacturers (S4 Table). The two studies (designated as study report no. 1 and report no. 2) also examined drug adsorption effects and device functionality [23]. The compatibility protocol of report no. 1 had an extended storage condition with etoposide, paclitaxel, methotrexate, and eight other drugs. Solutions were diluted to three times the drug's therapeutic level (concentration unspecified), using the vial spikes and closed male luer, and infused through administration sets. The closed male luer were stored for 120 days under refrigeration, followed by storage at room temperature for 7 days. The vial spikes were stored for 30 days under refrigeration and at room temperature for 7 days. A 24-h simulated infusion was tested for each drug. Due to the extended storage conditions, only cisplatin, etoposide, fluorouracil, and trastuzumab reported stable results. Moreover, the compatibility conditions were reported for extreme cases, and it was not possible to draw conclusions from report no. 1 regarding how the devices would behave under clinical conditions. Furthermore, the reasons behind drug selection criteria were not specified in report no. 1, and the dilution protocol did not reflect the actual clinical use. Busulfan, which are known to be incompatible, were not tested in report no. 1. However, report no. 1 included monoclonal antibodies (bevacizumab, cetuximab, and trastuzumab) as test drugs, which should become important from now on due to the increased use of biologics.

In report no. 2, etoposide (0.4 mg/mL), paclitaxel (1.2 mg/mL), and busulfan (0.54 mg/mL) were prepared at concentration levels similar to ours, which reflected clinical use concentrations. Syringe units were filled with undiluted drugs and stored for 24 h. The diluted solution was infused through the administration sets at 0 h and 24 h. Report no. 2 compared absorption peaks between solutions compounded using conventional syringes and CSTD. Differences in absorption peaks were used to indicate drug stability and chemical leakage from devices to the solution. Nevertheless, the experimental design had some deviations from the actual clinical usage. For instance, in this study, drugs were infused for 24 h. However, different drugs have different infusion times, such as paclitaxel, which is clinically infused over 3 h.

In contrast to the two studies, this present study measured drug potency instead of drug stability with the objective to determine the effect of CSTDs on drug potency, as per USP Chapter <800>, which states that antineoplastic drug concentrations must be verified after preparation, storage, and administration to ensure accurate therapeutic effects [2]. Most importantly, our results of the compatibility study are sensitive to study design, drug concentrations, and storage duration. These evidences of incompatibility between CSTD and antineoplastic drugs have justified the purpose of this study: to determine a suitable protocol for testing the underlying interactions between CSTD and antineoplastic drugs.

## Study limitations

Each test was performed in triplicate, which were all done by the same pharmacist on the same day. In a clinically simulated condition, the triplicate was preferrable to being done by different pharmacists and on different dates.

## Conclusion

The present study demonstrated the clinical applicability of simulated compatibility testing for CSTDs and antineoplastic drugs. We added clinical aspects to reflect actual clinical usage that CSTD incompatibility risks can be avoided by evaluating compatibility using a clinically simulated protocol. The maintenance of drug potency test was the most critical aspect while considering treatment effectiveness and toxicities. This study demonstrated a valuable methodology from clinicians' perspectives, especially for a drug that is relatively new and not tested by the manufacturers.

## Supporting information

**S1 Table. Drug selection.**
(DOCX)

**S2 Table. Drug preparations.**
(DOCX)

**S3 Table. HPLC conditions for analysis of antineoplastic drugs.**
(DOCX)

**S4 Table. Comparison of the results of two CSTD compatibility studies and the present study.**
(DOCX)

## Author Contributions

**Conceptualization:** Shao-Chin Chiang.

**Data curation:** Mandy Shen, Chen-Chia Lin.

**Formal analysis:** Mandy Shen, Chen-Chia Lin.

**Methodology:** Hui-Ping Chang.

**Project administration:** Shao-Chin Chiang.

**Supervision:** Chen-Chia Lin.

**Writing – original draft:** Mandy Shen.

**Writing – review & editing:** Shao-Chin Chiang.

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
