## [Decision Letter · Decision Letter 0]

24 Mar 2021

PONE-D-20-34542

Establishing a Protocol for the Compatibilities of Closed-System Transfer Devices with Multiple Chemotherapy Drugs under Simulated Clinical Conditions

PLOS ONE

Dear Dr. Chiang,

Thank you for submitting your manuscript to PLOS ONE. After careful consideration, we feel that it has merit but does not fully meet PLOS ONE’s publication criteria as it currently stands. Therefore, we invite you to submit a revised version of the manuscript that addresses the points raised during the review process.

It is recommended to to revised the manuscript based on the reviewers comment.

We look forward to receiving your revised manuscript.

Kind regards,

Girish Sailor

Academic Editor

PLOS ONE

Journal Requirements:

"This work was supported by Needleless Corporation (http://www.needleless-medical.com/en), which funded all costs associated with antineoplastic drugs and device purchase.  The funders had no role in study design, data collection and analysis, decision to publish, or preparation of the manuscript. No grant number was attached to this study."

We note that you received funding from a commercial source: Needleless Corporation.

Reviewers' comments:

Reviewer's Responses to Questions

**Comments to the Author**

1. Is the manuscript technically sound, and do the data support the conclusions?

Reviewer #1: Yes

Reviewer #2: Yes

2. Has the statistical analysis been performed appropriately and rigorously? 

Reviewer #1: N/A

Reviewer #2: Yes

3. Have the authors made all data underlying the findings in their manuscript fully available?

Reviewer #1: Yes

Reviewer #2: Yes

4. Is the manuscript presented in an intelligible fashion and written in standard English?

Reviewer #1: Yes

Reviewer #2: Yes

5. Review Comments to the Author

Reviewer #1: The manuscript reports an important aspect, which has not researched much. However, certain clarifications/modifications are required.

INTRODUCTION

1. ONB - spell in full first before being used as an abbreviation.

2. With reference to "manufacturer's claim: stated in Line 46, it is suggested to give a brief details of such claims, relevant to this study.

MATERIALS AND METHODS

1. Where application, give the references for the method adopted in the work.

2. Line 120, why half of the total dosage required to prepare one intravenous infusion bag was used? A brief clarification is required.

3. Line 176: It is suggested to give the source for the specifications.

4. Lines 182, 186, 195, 202 and 205 (or where applicable): It is suggested to enlist the materials used under the heading of Materials. They should not be mingled with the methods.

5. There is a need for a brief description of "simulated clinical conditions" in Materials and Methods.

CONCLUSION

1. There is a need to put the heading of the conclusion.

2. A sentence on the "maintenance of efficacy" should be a part of conclusion.

Reviewer #2: This reviewer has only minor comment.

Each test was performed in triplicate (page 5, line 102), but appeared to have been done by the same pharmacist (page 6, line 113). They should be done by different pharmacists, preferably on different days. This is the limitation of this study.

6. PLOS authors have the option to publish the peer review history of their article (what does this mean?). If published, this will include your full peer review and any attached files.

Reviewer #1: No

Reviewer #2: No

---

## [Author Response · Author response to Decision Letter 0]

30 Aug 2021

Reviewer #1: 

INTRODUCTION

1. ONB - spell in full first before being used as an abbreviation.

ONB is the product code for the product classification of Closed Antineoplastic and Hazardous Drug Reconstitution and Transfer System by the FDA. It is not an abbreviation.

2. With reference to "manufacturer's claim: stated in Line 46, it is suggested to give a brief detail of such claims, relevant to this study.

We have made modifications, please refer to Line 48 in the revised manuscript. 

MATERIALS AND METHODS

1. Where application, give the references for the method adopted in the work.

We do not have further reference material to provide. 

2. Line 120, why half of the total dosage required to prepare one intravenous infusion bag was used? A brief clarification is required. 

It has been clarified, please refer to Line 138-139 in the manuscript. 

3. Line 176: It is suggested to give the source for the specifications.

We have provided an additional reference for the source. Please see reference 21. 

4. Lines 182, 186, 195, 202 and 205 (or where applicable): It is suggested to enlist the materials used under the heading of Materials. They should not be mingled with the methods. 

The listed materials have been moved to appropriate sections. 

5. There is a need for a brief description of "simulated clinical conditions" in Materials and Methods.

This has been addressed please refer to Line 127.

CONCLUSION

1. There is a need to put the heading of the conclusion.

The heading has been added.

2. A sentence on the "maintenance of efficacy" should be a part of conclusion.

This has been addressed in the conclusion. Please see the revised version. 

Reviewer #2: 

Each test was performed in triplicate (page 5, line 102), but appeared to have been done by the same pharmacist (page 6, line 113). They should be done by different pharmacists, preferably on different days. This is the limitation of this study.

We had included a “study limitation” section within the discussion, please refer to Line 361.

---

## [Decision Letter · Decision Letter 1]

14 Sep 2021

Establishing a Protocol for the Compatibilities of Closed-System Transfer Devices with Multiple Chemotherapy Drugs under Simulated Clinical Conditions

PONE-D-20-34542R1

Dear Dr. Chiang,

We’re pleased to inform you that your manuscript has been judged scientifically suitable for publication and will be formally accepted for publication once it meets all outstanding technical requirements.

Kind regards,

Girish Sailor

Academic Editor

PLOS ONE

Additional Editor Comments (optional):

Reviewers' comments:

Reviewer's Responses to Questions

**Comments to the Author**

1. If the authors have adequately addressed your comments raised in a previous round of review and you feel that this manuscript is now acceptable for publication, you may indicate that here to bypass the “Comments to the Author” section, enter your conflict of interest statement in the “Confidential to Editor” section, and submit your "Accept" recommendation.

Reviewer #1: All comments have been addressed

Reviewer #2: All comments have been addressed

2. Is the manuscript technically sound, and do the data support the conclusions?

Reviewer #1: Yes

Reviewer #2: Yes

3. Has the statistical analysis been performed appropriately and rigorously? 

Reviewer #1: N/A

Reviewer #2: Yes

4. Have the authors made all data underlying the findings in their manuscript fully available?

Reviewer #1: Yes

Reviewer #2: Yes

5. Is the manuscript presented in an intelligible fashion and written in standard English?

Reviewer #1: Yes

Reviewer #2: Yes

6. Review Comments to the Author

Reviewer #1: (No Response)

Reviewer #2: The authors have adequately addressed the comments made during the initial review. I believe that this type of research should be encouraged.

7. PLOS authors have the option to publish the peer review history of their article (what does this mean?). If published, this will include your full peer review and any attached files.

Reviewer #1: No

Reviewer #2: No

---

## [Editor Report · Acceptance letter]

20 Sep 2021

PONE-D-20-34542R1 

Establishing a Protocol for the Compatibilities of Closed-System Transfer Devices with Multiple Chemotherapy Drugs under Simulated Clinical Conditions 

Dear Dr. Chiang:

I'm pleased to inform you that your manuscript has been deemed suitable for publication in PLOS ONE. Congratulations! Your manuscript is now with our production department. 

Kind regards, 

on behalf of

Dr. Girish Sailor 

Academic Editor

PLOS ONE